# Regression-Based Machine Learning for Predicting Lifting Movement Pattern Change in People with Low Back Pain

**DOI:** 10.3390/s24041337

**Published:** 2024-02-19

**Authors:** Trung C. Phan, Adrian Pranata, Joshua Farragher, Adam Bryant, Hung T. Nguyen, Rifai Chai

**Affiliations:** 1School of Science, Computing and Engineering Technologies, Swinburne University of Technology, Hawthorn, VIC 3122, Australia; tcphan@swin.edu.au (T.C.P.); adrian.pranata@rmit.edu.au (A.P.); hungnguyen@swin.edu.au (H.T.N.); 2School of Health Sciences, Swinburne University of Technology, Hawthorn, VIC 3122, Australia; 3College of Rehabilitation Sciences, Shanghai University of Medicine and Health Sciences, Shanghai 201318, China; joshua.farragher@rmit.edu.au; 4School of Health and Biomedical Sciences, RMIT University, Melbourne, VIC 3000, Australia; 5Centre for Health, Exercise and Sports Medicine, Department of Physiotherapy, The University of Melbourne, Melbourne, VIC 3010, Australia; albryant@unimelb.edu.au

**Keywords:** low back pain, lifting technique, camera system, sagittal plane, trunk, hip, knee, range of motion, regression machine learning, forecast

## Abstract

Machine learning (ML) algorithms are crucial within the realm of healthcare applications. However, a comprehensive assessment of the effectiveness of regression algorithms in predicting alterations in lifting movement patterns has not been conducted. This research represents a pilot investigation using regression-based machine learning techniques to forecast alterations in trunk, hip, and knee movements subsequent to a 12-week strength training for people who have low back pain (LBP). The system uses a feature extraction algorithm to calculate the range of motion in the sagittal plane for the knee, trunk, and hip and 12 different regression machine learning algorithms. The results show that Ensemble Tree with LSBoost demonstrated the utmost accuracy in prognosticating trunk movement. Meanwhile, the Ensemble Tree approach, specifically LSBoost, exhibited the highest predictive precision for hip movement. The Gaussian regression with the kernel chosen as exponential returned the highest prediction accuracy for knee movement. These regression models hold the potential to significantly enhance the precision of visualisation of the treatment output for individuals afflicted with LBP.

## 1. Introduction

Low back pain (LBP) is a common and debilitating condition affecting millions worldwide. Activities of daily living, such as repetitive lifting, have been associated with LBP. Lifting is an intricate task that necessitates coordination of the lower limbs (such as the hip and knee) as well as the trunk [1]. Poor lifting mechanics can occur for various reasons, such as lifting objects that are too heavy, lifting an object from an inappropriate height, lifting awkwardly shaped objects, or performing repetitive lifting tasks without proper rest and recovery [2,3,4]. Therefore, understanding and monitoring the changes in lifting movement could be critical for effective rehabilitation and prevention of work-related LBP.

LBP is also associated with changes in the lumbar spine, hip, and knee movements. Current treatment options for LBP typically involve physical therapy. The common therapy incorporates general strength training and aims to restore the baseline function of people with LBP [5,6]. However, due to the absence of advanced technology in clinical practice, clinicians typically rely on visual observation and patient questionnaires. These surveys inquire about the level of pain experienced by patients and the functional activities they can perform but provide little to no information on how the task (e.g., lifting task) is performed.

Recent technological advancements, such as motion capture systems and machine learning algorithms, have shown great potential in objectively identifying and correcting movement patterns during lifting tasks or patient flow [7,8]. Motion capture systems can capture detailed movement data during lifting tasks, including joint angles and velocity. Machine learning algorithms can then be trained on these data to identify abnormal movement patterns and provide personalised corrective feedback to the individual. There are two main applications of machine learning: classification and regression. Recent research has presented the potential of machine learning in clustering lifting movements in people with LBP and healthy people [7] or classifying thyroid disease [9]. Regression analysis is a commonly used statistical technique for modelling the associations among variables. It entails forecasting a continuous output value by considering one or more input variables. In recent years, with the rise of big data and advancements in machine learning techniques, regression analysis has become a critical tool in various fields, including energy, finance, healthcare, and marketing [10,11].

Regression machine learning algorithms are artificial intelligence that uses historical data to learn from patterns and make predictions about future outcomes. These algorithms can analyse large datasets, identify complex patterns, and predict continuous output values with high accuracy. Currently, many regression machine learning models have been developed. Regression machine learning has several advantages over traditional statistical regression methods, including handling nonlinear relationships between variables and incorporating multiple variables and interactions.

In recent years, regression machine learning algorithms have experienced rapid growth due to their demonstrated high accuracy across a range of applications. For instance, regression machine learning has been used to predict stock prices [12], diagnose diseases [13], forecast weather patterns [14,15], and assess molecular similarity [16]. In addition, regression machine learning has been used to optimise processes and improve decision making in industries such as manufacturing and transportation [17,18,19,20]. Some studies have suggested that machine learning algorithms, particularly regression models, may have the potential to predict treatment outcomes based on patient characteristics and movement patterns [21]. However, this technique has not been applied to analyse lifting movement in people with LBP.

Hence, the purpose of this study is to discover the potential application of regression machine learning models for predicting alterations in joint movement during lifting tasks following a 12-week general strength treatment for individuals with LBP. The output model can be applied to provide insights into the potential achieved range of motion of a specific body segment after 12 weeks of strength training. We hypothesise that our pilot study will enhance comprehension regarding the potential of machine learning in forecasting treatment outcomes for patients with LBP. Furthermore, it could pave the way for more effective methods for visualising treatment outputs.

Recently, we conducted a research study applying machine learning to cluster joint movements during lifting tasks based on the ROM of the sagittal plane in the trunk, hip, and knee [7]. Our results showed that the Ward clustering method successfully identified four distinct joint movement patterns. However, while clustering is a core application of machine learning, its use with joint movement data for regression tasks remains unexplored. Nevertheless, a separate study compared existing regression algorithms for predicting brain age and yielded promising results [22]. This suggests the potential for applying the existing regression approaches to joint movement data.

The core contribution of this paper is the innovative use of regression machine learning methods to predict lifting movement patterns in participants with LBP following a 12-week general strength treatment. This study is the first to use information on the range of motion for the trunk, hip, and knee, along with a regression algorithm, to anticipate changes in movement patterns in the sagittal plane.

## 2. Materials and Methods

### 2.1. Participants

Sixty-nine participants, both males and females aged between 18 and 65 years old (falling within the “adult” age range [23]) and experiencing lower back pain (33 of whom were female), were enlisted from a prominent Physiotherapy clinic in Melbourne, Victoria, Australia. Approval for this study was obtained from the University of Melbourne Behavioural and Social Sciences Human Ethics Sub-Committee. Inclusion criteria comprised individuals reporting pain between the gluteal fold and the twelfth thoracic vertebra (T12) level, with or without leg pain persisting for more than three months. Exclusion criteria encompassed the presence of evident neurological signs, for example, muscle weakness and loss of lower limb reflexes, a history of spine and lower limb surgery, diagnosis of active inflammatory conditions like rheumatoid arthritis, a cancer diagnosis, or a lack of proficiency in written or verbal English. All participants underwent evaluations of pain using the pain self-efficacy questionnaire (PSEQ) [24]. Participants were recruited and received strengthening exercise treatment for 12 weeks. During the 12 weeks of treatment, participants joined exercise sessions twice per week. The assessments, in which participants were asked to perform lifting tasks, were conducted on the first week, week 6, and week 12.

### 2.2. Data Collection

The preceding investigation outlined the lifting task protocol [25,26]. Participants started in a standing position, barefoot, with their arms alongside their bodies. They were required to bend down as directed and perform lifting tasks with an 8 kg weight (equivalent to the average weight of groceries [27]). The weight was placed between their feet. The weight was lifted from the ground to their belly by using both hands. A lifting technique of their preference was allowed to be used without any restriction. The first and second lifting tasks were practice trials and, consequently, excluded from further analysis. The participants were instructed to repeat the lifting task six times.

Kinematic data were gathered by affixing non-reflective markers to specific anatomical points on the participants’ skin, including the head, trunk, pelvis, and upper and lower limbs [26]. A motion analysis system consisting of 12 cameras (Optitrack Flex 13, NaturalPoint, Corvallis, OR, USA) with a sampling rate of 120 Hz was applied to create three-dimensional recordings of anatomical reference point. Optitrack Motive software v2.0 (NaturalPoint, Corvallis, OR, USA) was used to process kinematic data with grouping, naming, cleaning, and gap-filling. Following this, a pipeline with some modifications was used for further processing using in Visual3D v5.01.6 (C-Motion, Inc., Germantown, MD, USA) to extract the velocity and angular data of various joints in all planes. An overview of the data collection process is summarised in Figure 1.

### 2.3. Pre-Processing and Feature Extraction

The analysis involved utilising the angular rotation data from the three different joints (trunk, hip, and knee) throughout the lifting process, which were used as the input for the machine learning algorithm. This study selected a range of motion (ROM) of different body segments to transform the complicated information into more manageable features. This ROM was determined by computing the variance between the maximum and minimum values of the rotational displacement for each respective joint as follows: (1)$$Range\;of\;motion\;\left(ROM\right)=Max\left(\partial \right)-Min(\partial )$$
where $Max\left(\partial \right)$ represents the maximum of the rotational displacement for the joint and $Min\left(\partial \right)$ denotes minimum of the rotational displacement for the body segment.

This view of inter-joint coordination during manual lifting proposes a sequence extending from peripheral (further from the centre of the body) to central (closer to the centre) for the vertebral joints of the knee, hip, and belt [28]. Furthermore, the motion of the knee, hip, and lumbar areas is essential for completing the lifting task and performing diverse lifting techniques. The processed data focused on extracting the ROM in the sagittal plane for the trunk, hip, and knee, which was used for further analysis. The knee and hip each used an average value between sides as no statistically significant differences in the ROM were detected between the right and left sides.

### 2.4. Regression Machine Learning

Regression machine learning serves as a powerful instrument for forecasting continuous values using input features. It includes instructing a model using a dataset containing known input–output pairs and subsequently utilising the trained model to forecast the output for new input data. Regression machine learning tries to discovery a mathematical function in which the input features are mapped and predicted to the output values, such that the predicted values are as close as possible to the true values. 

This study used three different regression models to predict changes in trunk, hip, and knee ROM over a 12-week treatment period, with predictions made every 6 weeks. The input for the models consisted of a combination of trunk, hip, and knee ROM measurements taken in the first and sixth weeks of treatment. The actual outputs used to evaluate the models were the ROM measurements for the trunk, hip, or knee in the sixth week for the first-week input and in the twelfth week for the sixth-week input, depending on which regression model was used. The input was normalised before use as the input for the regression model. 

The regression algorithms that were assessed in this study for predicting the change in trunk, hip, and knee movement are explained below.

#### 2.4.1. Supported Vector Machine Regression

Support Vector Regression (SVR) is a machine learning algorithm suggested in line with the Support Vector concept that was initially introduced [29,30]. SVR, as a form of a supervised learning algorithm, aims to reduce the discrepancy between the forecasted values and the true labels. This is achieved by identifying a hyperplane that effectively divides the data into distinct classes. In contrast to traditional regression methods, in which the squared error between the forecasted and true labels is reduced, with SVR, the range between forecasted values and true labels is minimised. This makes it a more robust algorithm, as it is not as sensitive to outliers in the data.

A primary benefit of SVR is its flexibility in dealing with complex geometries and the transmission of data. This means that it can be used effectively even in cases where the data are highly nonlinear or where there is noise in the data. Additionally, SVR provides additional kernel functionality, in which the model’s capability is enhanced for forecasts by reflecting the characteristics of features. The kernel functionality of SVR is one of its most significant strengths, as it allows the algorithm to convert the input data into a space with higher dimensions, making the data more readily distinguishable.

#### 2.4.2. Binary Decision Tree Regression

One type of supervised machine learning method involving a series of binary decisions based on attributes is known as a Binary Decision Tree [31]. Every determination results in one of two potential outcomes: it either leads to another determination or culminates in a forecast. Using each independent variable, the model fits the target variable in a regression tree. The next step involves dividing the data into groups based on different values of the independent variables. At each point, the difference between the predicted and actual values is squared to calculate the “Sum of Squared Errors” (SSE). By comparing the SSE across all variables, the potential separated point will be selected at the point has the lowest SSE value. This process recurs and continues until the final output value is predicted.

#### 2.4.3. Ensemble Tree Regression

Ensemble learning utilises the strengths of multiple weak learners and produces models with slightly better performance than random chance. This helps in building a strong learner with significantly enhanced predictive performance [32,33]. This approach often leads to better performance than using individual learners. One common form of ensemble learning is ensemble trees, which combine the forecasts of multiple decision trees in order to generate significantly more accurate prognostic information compared with a single decision tree. The key principle behind ensemble trees is that a strong learner is formed from the collective strength of multiple weak learners.

Several techniques are operated to function ensemble trees, including bagging and least-squares boosting. Bagging is used with the main goal of decreasing the discrepancy in a decision tree. This process involves randomly drawing data points from the original dataset with replacement, producing multiple subsets [34]. These subsets play an important role in training a decision tree, leading to the creation of an ensemble of diverse models. The final forecast is obtained by averaging the forecasts from each individual tree in the ensemble, resulting in a more robust forecast compared to relying solely on a single decision tree. On the other hand, in least-squares boosting (LSBoost),regression ensembles are determined by optimising the fitting of a new regression model at each step based on the dissimilarity between the observed outcome and the current ensemble’s forecast [22]. The current ensemble’s forecast is generated by combining the forecasts of all previously grown learners. The final step involves adjusting the ensemble to decrease the overall error in its forecasts, measured by the mean squared error. This approach is particularly effective for regression problems.

#### 2.4.4. Gaussian Processes for Regression

A machine learning algorithm specifically designed for regression analysis tasks is Gaussian Process regression [35]. In contrast to other regression methods that estimate the parameters of a specific function, Gaussian Process regression distinguishes itself by its ability to calculate the probability distribution over all possible functions, providing a more flexible and data-driven approach to modelling complex relationships. For Gaussian processes, there is a wide variety of available kernel functions including the following:

Squared Exponential Kernel:(2)$$k\left(a_{i},a_{j}|\beta \right)=\tau_{f}^{2}exp\left[-\frac{1}{2}\frac{{\left(\alpha_{i}-\alpha_{j}\right)}^{T}(\alpha_{i}-\alpha_{j})\;}{\tau_{l}^{2}}\right]$$
where $a_{i}$ and $a_{j}$ are n-dimensional input vectors, β represents the kernel parameters, $\tau_{f}^{2}$ is the signal standard deviation, which controls the overall scale of the function’s output, $\tau_{l}^{2}$ is characteristic length scale, which controls the smoothness and influence of distant points, and ${\left(\alpha_{i}-\alpha_{j}\right)}^{2}(\alpha_{i}-\alpha_{j})$ is squared Euclidean distance between $a_{i}$ and $a_{j}$.

Exponential Kernel:(3)$$k\left(a_{i},a_{j}|\beta \right)=\tau_{f}^{2}exp\left(-\frac{c\;}{\tau_{l}}\right)\;\;\;\;\;\;\;$$
where
(4)$$c=\sqrt{{\left(\alpha_{i}-\alpha_{j}\right)}^{2}(\alpha_{i}-\alpha_{j})}$$


Matern 3/2:

(5)
$$k\left(a_{i},a_{j}|\beta \right)=\tau_{f}^{2}\left(1+\frac{\sqrt{3}\;c}{\tau_{l}}\right)exp\;\left(-\frac{\sqrt{3}\;c\;}{\tau_{l}}\right)\;\;\;$$




Matern 5/2:

(6)
$$k\left(a_{i},a_{j}|\beta \right)=\tau_{f}^{2}\left(1+\frac{\sqrt{5}\;c}{\tau_{l}}+\frac{5\;c^{2}}{3\;\tau_{l}^{2}}\right)exp\left(-\frac{\sqrt{5}\;c\;}{\tau_{l}}\right)$$



Rational Quadratic Kernel:(7)$$k\left(a_{i},a_{j}|\beta \right)=\tau_{f}^{2}{\left(1+\frac{c^{2}\;}{2\gamma \tau_{l}^{2}}\right)}^{-\gamma }\;\;\;\;\;\;\;\;\;\;\;\;\;\;$$
where γ represents a positive-valued scale-mixture parameter. 

Gaussian Process regression is a non-parametric regression method, meaning it makes no assumptions about the shape or form of the underlying function. Instead, the relationship between the input and output variables is modelled as a distribution of functions.

#### 2.4.5. Linear Regression

Linear regression is a parametric statistical method for modelling the linear relationship between a single continuous dependent variable and one or more independent variables, also known as explanatory variables [36]. This approach involves constructing a linear predictor function, which estimates the dependent variable’s value based on the independent variables’ values. This method aims to find the straight line that best represents the data points, revealing the underlying relationship between the variables. This relationship is explained by the linear predictor function in a mathematical formula [36]. This function is represented as a straight line in a two-dimensional graph, where the variable whose value is predicted (dependent variable) is positioned on the *y*-axis, while the variable(s) used for prediction (independent variable(s)) are positioned on the *x*-axis. The strength of the relationship is represented by the slope of the line, and the y-intercept clarifies what the dependent variable would be if the independent variable(s) were zero.

Linear regression models can be fitted using a variety of approaches, but the most common method is the least-squares approach. This method aims to find the best fit by minimising the total error between the forecasted and real values of the dependent variable.

### 2.5. Performance Evaluation

The predictive performance of various algorithms for estimating the ROM of the trunk, hip, and knee was assessed using a 10-fold cross-validation approach on the training data. This technique involved dividing the training set into 10 equal parts, training each algorithm on a combination of 9 parts, and evaluating its prediction accuracy on the remaining part. Performance analysis for a regression model involves evaluating the accuracy and reliability of the model’s predictions. There are a few common ways to evaluate a regression model: mean absolute error, R^2^, and root mean squared error.

#### 2.5.1. Mean Absolute Error (MAE)

A popular metric for assessing a regression model’s performance is the mean absolute error (MAE). This metric measures and calculates the average of the absolute dissimilarity between the outcomes forecasted by the regression machine learning and the actual observation.
(8)$$MAR=\frac{1}{m}\times \sum \left|y_{pred}-y_{true}\right|$$
where $y_{pred}$ is the forecasted output, $y_{true}$ is the real observation, m is the number of observations, and Σ is the sum of all observations.

The MAE shows how far off the predictions are on average. It is useful for models where the absolute error is more important than the squared error and is not sensitive to outliers, in contrast to the RMSE. A lower MAE indicates a finer fit of the model to the data, meaning the forecasted outcomes are closer to the real observations on average.

#### 2.5.2. R-Squared (R^2^)

In regression analysis, R^2^, or the coefficient of determination, is a key metric implemented to measure the percentage of the discrepancy in the dependent variable, which can be justified by the independent variables. This provides valuable insights into the model’s capability to capture the connection between the input and output variables. R^2^ values, ranging from 0 to 1, represent the proportion of the explained discrepancy to the total discrepancy in the dependent variable in a regression model. A value of 1 signifies a perfect fit, meaning the independent variables completely explain the discrepancy in the dependent variable, while a value of 0 signals that the regression machine learning offers no explanatory power beyond the mean. This is commonly explained as the percentage of the sum of the variation in the dependent variable that the model explains. R^2^ is often used as a performance metric for regression models, with higher values indicating better model performance.

#### 2.5.3. Root Mean Squared Error (RMSE)

Root mean squared error (RMSE) is another performance metric frequently utilised in regression machine learning tasks to assess the correctness of a model. It achieves this by first determining the squared difference between each predicted value and its corresponding actual value, averaging these squared differences, and then taking the square root of the mean.
(9)$$MAR=\sqrt{\frac{1}{m}\times \sum (y_{pred}-y_{true}{)}^{2}}$$
where $y_{true}$ represents the real observations, $y_{pred}\;$ represents the forecasted outputs, and m is the number of observations.

The RMSE measures the average magnitude of forecast errors, with lower values signifying better performance. This metric uses the same units as the target variable, enabling straightforward interpretation and comparison of different model capabilities.

## 3. Results

Eight-hundred and sixty-four data points were included in this study. This dataset was broken into two sets: the training set (*n* = 692) and the testing set (*n* = 172). The demographics of the study participants are summarised in Table 1 [26]. 

Figure 2, Figure 3 and Figure 4 present a detailed comparison of the performance achieved by different forecast algorithms for trunk, hip, and knee movements in the training dataset.

For the trunk, hip, and knee regression model, based on the R^2^, for the training set, Linear SVR, Polynomial SVR, and linear regression provided a low coefficient of determination R^2^ (<0.6) while the other regression models presented a high coefficient of determination R^2^. Hence, these regression models were not suitable for predicting the change in trunk, hip, and knee movement. 

The Ensemble Tree model (LSBoost) exhibited the optimal estimation accuracy in the trunk regression task. This was evidenced by its significantly lower MAE (1.24 degrees) and RMSE (1.95 degrees), coupled with its high R^2^ value (0.97) when compared with the other models. In the training set, this superior performance could be attributed to the model’s ability to achieve an almost precise fit to the data points. Linear SVR on the training dataset showed significantly inferior performance compared with the other prediction models, as evidenced by its considerably higher MAE (8.38 degrees) and RMSE (10.46 degrees) and substantially lower R^2^ (0.11).

Similar to the trunk regression task, Ensemble Tree (LSBoost) exhibited the optimal estimation accuracy in the hip regression task. This was demonstrated by its considerably lower MAE (1.25 degrees) and RMSE (2.59 degrees), combined with its high R^2^ value (0.96). Linear SVR on the training dataset for the hip presented significantly lower performance compared with the other forecast regression model, as evidenced by its considerably higher MAE (9.01 degrees) and RMSE (11.94 degrees) and substantially lower R^2^ (0.20).

With the training set for the knee, Ensemble Tree (LSBoost) also demonstrated optimal prediction accuracy with its lower MAE (2.96 degrees) and RMSE (5.65 degrees), coupled with its high R^2^ value (0.96) compared with the other regression model. The Linear SVR model performed significantly worse than other models when applied to the training dataset. This was evident from its high MAE (17.79 degrees) and RMSE (25.59 degrees) values, indicating large average errors, and its low R^2^ value (0.13), signifying poor explanatory power.

The accuracy of several regression machine learning algorithms for forecasting trunk, hip, and knee movement in the test set is shown in Figure 5, Figure 6 and Figure 7.

It is noteworthy that most forecast regression models established remarkable accuracy in their forecasts (high R^2^ values and a mean of the ROM delta close to zero) except for Linear SVR, Polynomial SVR, and linear regression.

For the trunk regression model, the range of the MAE was from 2.05 to 8.14 degrees. Ensemble Tree (LSBoost) demonstrated the highest prediction accuracy in which the MAE was 2.05 degrees, RMSE was 2.99 degrees, and R^2^ was 0.92 for the test dataset. Evaluation of the testing set revealed that the performance of the Linear SVR algorithm fell short of other forecast algorithms. This was demonstrated by its higher MAE (8.13 degrees) and RMSE (10.37 degrees) values and its lower R^2^ (0.064).

For the hip regression model, the range of the MAE was from 1.955 to 9.23 degrees. Ensemble Tree (LSBoost) also presented the highest prediction accuracy in which the MAE was 1.95 degrees, RMSE was 3.09 degrees, and R^2^ was 0.94 for the test dataset. Linear SVR on the test dataset for the hip showed significantly lower performance compared with the other forecast regression model, as evidenced by its considerably higher MAE (8.12 degrees) and RMSE (10.74 degrees) and substantially lower R^2^ (0.26).

For the knee regression model, since the knee was more flexible (the ROM of the knee is much larger than that of the trunk and the hip), the range of MAE was higher than the trunk and hip, as expected (from 9.42 to 28.07 degrees). Gaussian regression with the kernel chosen as exponential provided the optimal estimation accuracy in the knee regression task with the test dataset. This was demonstrated by its significantly lower MAE (6.04 degrees) and RMSE (9.42 degrees), combined with its high R^2^ value (0.90) compared with the other model. Linear SVR displayed unsatisfactory performance on this testing set. The result for Linear SVR reported a high MAE and RMSE value and a low R^2^ value (MAE = 19.83 degrees, RMSE = 28.07 degrees, R^2^ = 0.109).

## 4. Discussion

The application of regression machine learning in healthcare has seen significant growth in recent years, with its use extending to various areas such as disease diagnosis, prognosis prediction, and treatment recommendation. A thorough review of the existing literature revealed that no prior research has investigated the application of regression machine learning specifically for estimating the lifting movements in people with LBP after a course of treatment. By leveraging the power of regression machine learning, which has demonstrated its efficacy in various healthcare domains, we aim to provide valuable insights into the predictive capabilities of these models for ROM changes in different joint segments. These models can be valuable tools for evaluating health status, identifying potential clinical issues, assessing the risk of musculoskeletal impairments in individuals, or offering clinicians and researchers a reliable tool for evaluating treatment outcomes and tailoring interventions to optimise patient outcomes. In contrast to the current technology, which lacks the ability to discern how movement during the lifting task may change after different training methods, the predicted values enable the clinic to potentially understand where the target range of motion can be achieved using various training methods. This guides clinicians in selecting the appropriate treatment for the patient. Motivated by the critical role of the trunk, hip, and knee ROM in lifting tasks, the primary objective of this study was to identify the most effective algorithms for forecasting these movement parameters in individuals with LBP following a course of treatment.

In this study, a total of twelve regression models, both linear and nonlinear, were assessed. In a previous research study, which focused on predicting brain age using various existing regression machine learning algorithms, the results indicated that the Quadratic Support Vector Regression algorithm performed the best, while the Binary Decision Tree algorithm provided the worst predictions [22]. In contrast, our research findings suggest that the Ensemble Tree (LSBoost) and Gaussian regression with Kernel (chosen as Exponential) returned the highest prediction accuracy for trunk, hip, and knee movements on the test set. Surprisingly, the Binary Decision Tree algorithm exhibited high accuracy in trunk, hip, and knee movements, in contrast to its performance in predicting brain age, where it yielded the lowest accuracy. These results suggest that the optimal choice of a regression algorithm can vary significantly depending on the specific application domain. For our study, the linear regression models examined were linear regression and Linear Support Vector Regression (SVR). On the other hand, the nonlinear regression methods encompass SVR with Polynomial and Gaussian kernels, Ensemble Trees, Binary Decision Tree, and Gaussian regression. The analysis of the regression models revealed that linear regression models had the highest error rate compared with the other methods. This outcome suggests that a linear relationship may not adequately capture the underlying trend in the data. Its weak performance implies that the relationship between these variables is likely more complex and nonlinear. Upon evaluating the various regression models, it was observed that both Gaussian SVR and Polynomial SVR yielded similarly poor results as linear regression. It is evident that the change in the trunk, hip, and knee ROM after a course of treatment does not conform to a simple linear or polynomial relationship. 

In the analysis of the trunk, hip, and knee models, it was observed that the Gaussian regression model consistently exhibited similar performance across different kernel functions. This implies that the choice of kernel function did not significantly impact the predictive capabilities of the Gaussian regression model for these particular models. The stable and consistent performance of the Gaussian regression model across various kernel functions suggests that it possesses inherent robustness and adaptability in capturing the underlying relationships between the predictor variables and the ROM outcomes for the trunk, hip, and knee. This finding highlights the versatility of the Gaussian regression model and its ability to provide reliable predictions regardless of the specific kernel function utilised. In some previous studies, Gaussian regression showed similar positive results in other applications, such as developing forecasts of upper limb rehabilitation success for brain injury survivors based on clinical and wearable sensor data [37] and predicting atomic energies and multipole moments [38].

The results of the regression models also revealed that the MAE for the knee model was higher than that for the trunk and hip models. Given the diverse age range of the participants, even though it falls within the “adult” age range, this variability could potentially be a correlation factor. However, we conducted a further analysis to explore whether age correlates with the high MAE for the knee model. Examining the MAE values from the best regression model for the knee in the testing set (Gaussian regression with the kernel chosen as exponential) and age, we identified that there appears to be no correlation between age and the MAE of the knee, as the MAE values were high across all age groups. Figure 8 outlines the MAE values for the knee regression model and the age of the participants.

The box plots for the ROM delta (the actual ROM subtracted from the predicted ROM) for the trunk, hip, and knee between the various regression models over the testing set are visualised in Figure 9, Figure 10 and Figure 11, respectively. The labels on the box plots are related to the serial numbers (S.Nos) of the regression machine learning listed in Figure 2, Figure 3, Figure 4, Figure 5, Figure 6 and Figure 7. The box plots show that the test set’s ROM delta is almost zero for the trunk, hip, and knee. However, the interquartile range (IQR) was slightly larger in the Linear SVR, Gaussian SVR, Polynomial SVR, Ensemble Tree (Bag), and linear regression models for the trunk, hip, and knee. 

Based on the box plots, we can observe the presence of outliers in both cases. For the best trunk model (Ensemble Tree (LSBoost)), the number of outliers is seven (around 4.1%). For the best hip model (Ensemble Tree (LSBoost)), the number of outliers is 16 (around 9.3%). For the best knee model (Gaussian regression (kernel—exponential)), the number of outliers is 14 (around 8.1%). These outliers can be attributed to the fact that participants transitioned to a completely different lifting technique after the 12 weeks of treatment, which significantly deviates from their previous lifting technique. On the other hand, this also means that after 12 weeks of treatments, the regression machine learning suggests that there is a 5% chance that participants will significantly change their movement pattern in the trunk, 10% in the trunk, and 9% in the knee. Additionally, regarding the regression for trunk, hip, and knee movements, the variation between the forecasted and real values is quite minimal, with a difference of less than 5 degrees. This implies that the regression model effectively forecasts the alteration of movements after 12 weeks of treatment in most situations. By combining the regression machine learning model for the trunk, hip and knee, a real-time prediction model can be constructed. The proposed structure of the real time model is presented in Figure 12.

One of the limitations of this study is that the experiment solely focused on measuring the trunk, hip, and knee ROM in the sagittal plane, neglecting movements in the coronal and axial planes. Although alternative approaches could have been considered, the chosen method remained appropriate as the symmetrical lifting task largely involved movement within the sagittal plane, focusing primarily on the lumbar spine, hip, and knee. Future research is recommended to investigate regression models that incorporate the ROM of these joints in all planes to gain a more comprehensive understanding. This innovative approach holds promise for guiding more informed assessments and targeted rehabilitation strategies for individuals with LBP. Future clinical trials are needed to fully validate its effectiveness in real-world settings. Authors should engage in a comprehensive discussion that examines the results in relation to the existing literature, the initial research hypotheses, and their broader implications for the field. This discussion should encompass the full scope of the findings and their potential applications, outlining promising directions for future research. Finally, the regression machine learning model failed to achieve 100% prediction accuracy for changes in movement patterns following the treatment in this study, suggesting potential limitations in its ability to perfectly capture the underlying relationships between variables. To gain a deeper understanding of this group and its unique characteristics, additional research should be undertaken. Exploring potential factors contributing to the unexplained variance could uncover valuable insights and help refine the predictive model for more accurate assessments in future studies. Alternatively, in future research, researchers can aim to explore the transition from the ROM of trunk, hip and knee data to image data extraction using a similar motion analysis system equipped with 12 cameras. Adopting this approach can harness the capabilities of deep learning models, specifically Convolutional Neural Networks (CNNs), for a more nuanced understanding of motion patterns. This shift holds the potential to elevate the precision and depth of the analysis, paving the way for enhanced insights into motion dynamics.

## 5. Conclusions

Based on our comprehensive examination of relevant scholarly publications, this research is the earliest pilot research exploration using regression machine learning to predict changes in trunk, hip, and knee movement after 12 weeks of strength training. To predict trunk movement, the Ensemble Tree (LSBoost) returned the highest prediction accuracy. The Ensemble Tree (LSBoost) returned the highest prediction accuracy for hip movement prediction. The Gaussian regression with the kernel chosen as exponential returned the highest prediction accuracy for knee movement. This innovative approach offers the potential for more precise evaluation and clearer visualisation of how treatment impacts patients with LBP.

## Figures and Tables

**Figure 1 sensors-24-01337-f001:**
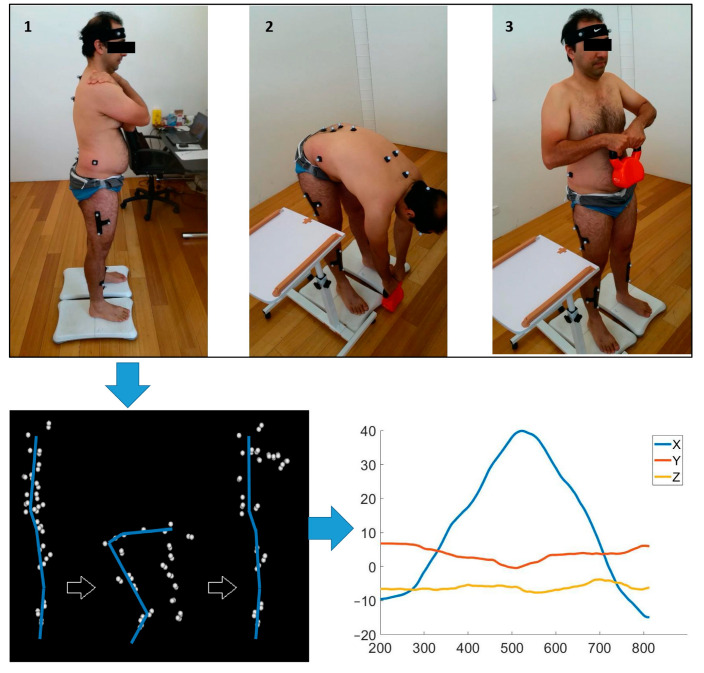
Overview of the data collection process.

**Figure 2 sensors-24-01337-f002:**
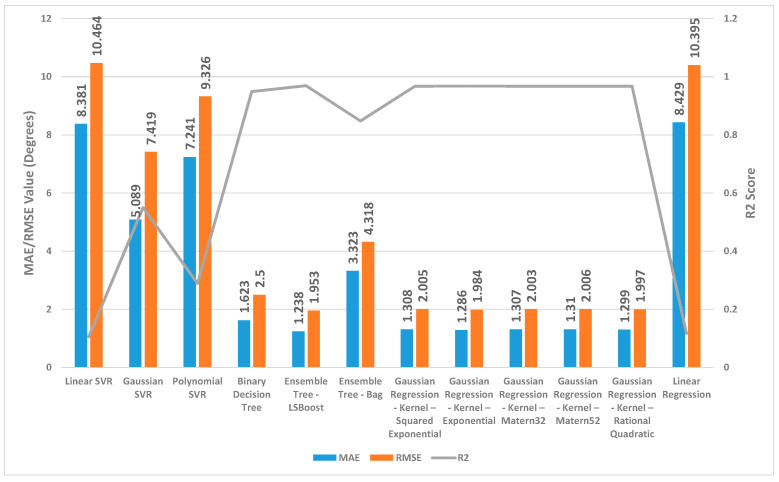
Summary of the performance results for several regression machine learning algorithms utilised to forecast trunk movement in the training set.

**Figure 3 sensors-24-01337-f003:**
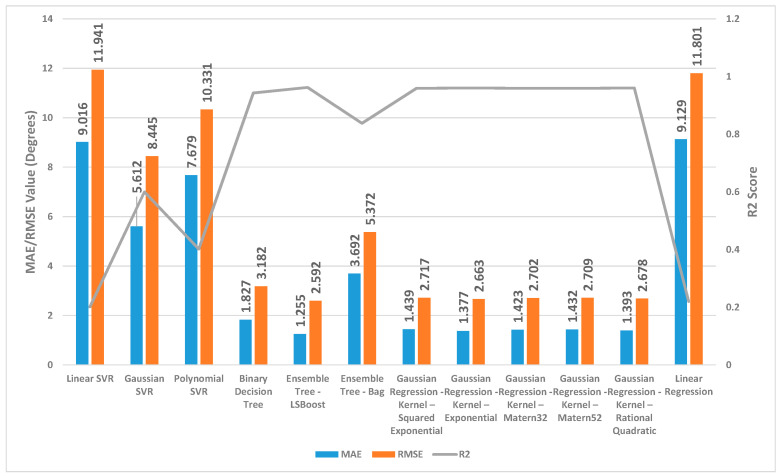
Summary of the performance results for several regression machine learning algorithms utilised to forecast hip movement in the training set.

**Figure 4 sensors-24-01337-f004:**
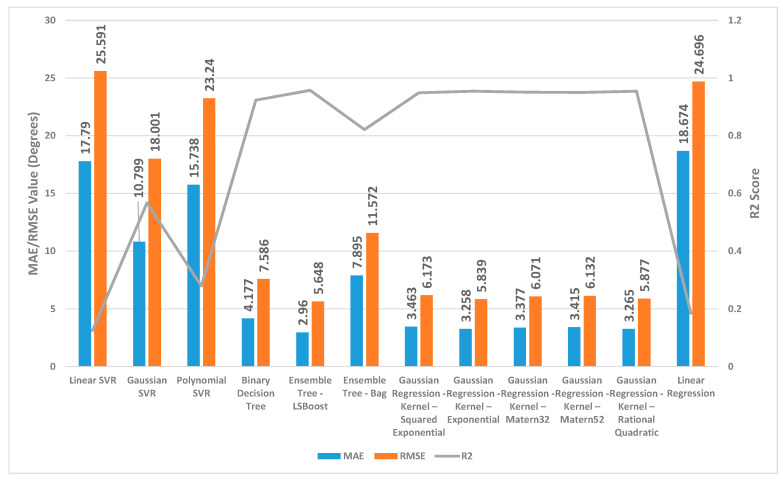
Summary of the performance results for several regression machine learning algorithms utilised to forecast knee movement in the training set.

**Figure 5 sensors-24-01337-f005:**
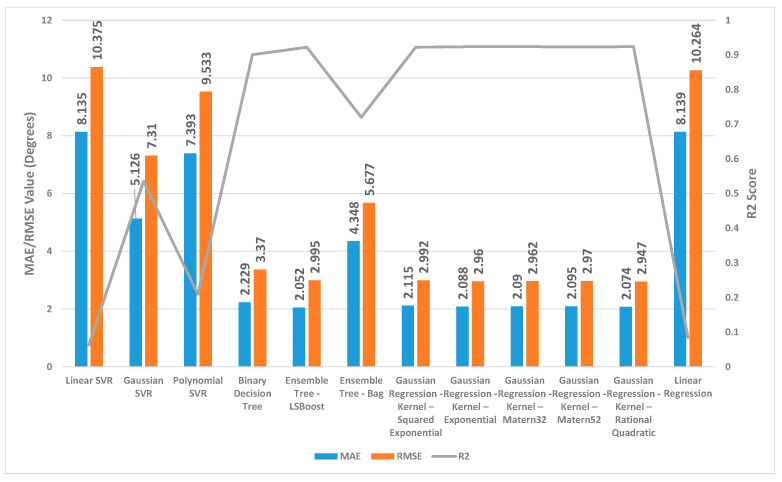
Summary of the performance results for several regression machine learning algorithms utilised to forecast trunk movement in the test set.

**Figure 6 sensors-24-01337-f006:**
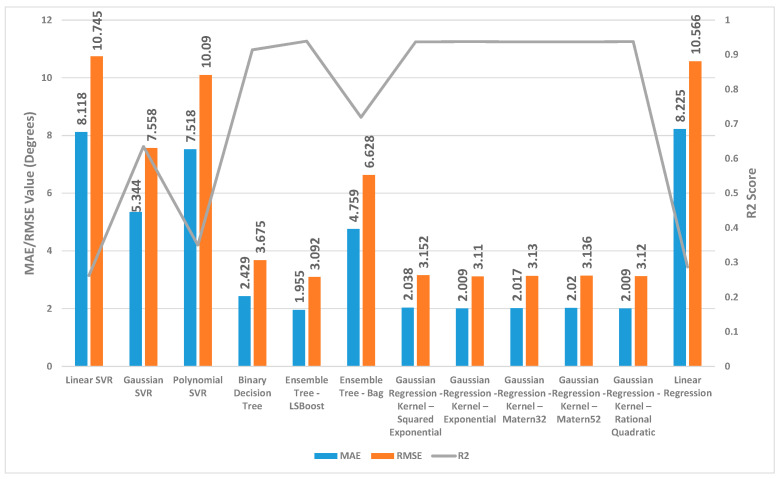
Summary of the performance results for several regression machine learning algorithms utilised to forecast hip movement in the test set.

**Figure 7 sensors-24-01337-f007:**
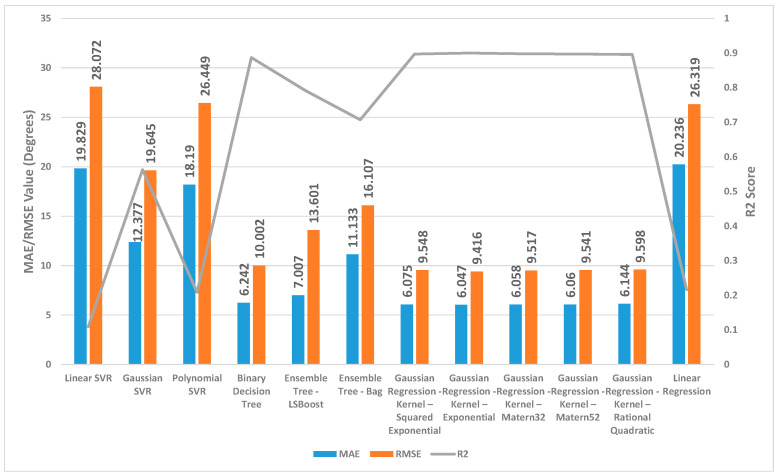
Summary of the performance results for several regression machine learning algorithms utilised to forecast knee movement in the test set.

**Figure 8 sensors-24-01337-f008:**
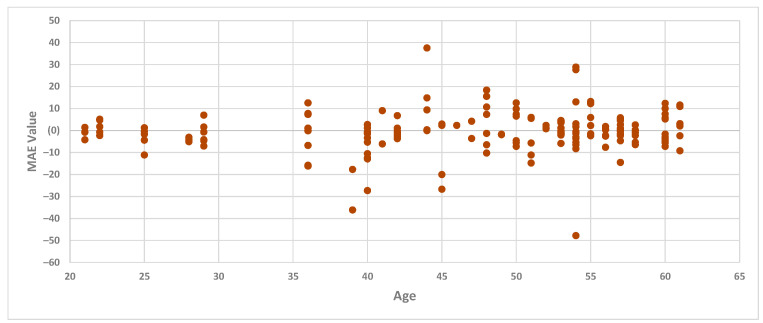
Summary of the comparison between the MAE values for the Gaussian regression with the kernel chosen as exponential and the age of participants.

**Figure 9 sensors-24-01337-f009:**
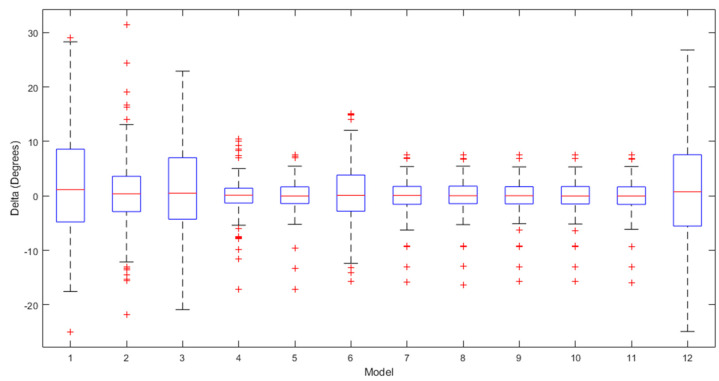
Box plots summarising the trunk ROM changes across various regression machine learning algorithms on the test set.

**Figure 10 sensors-24-01337-f010:**
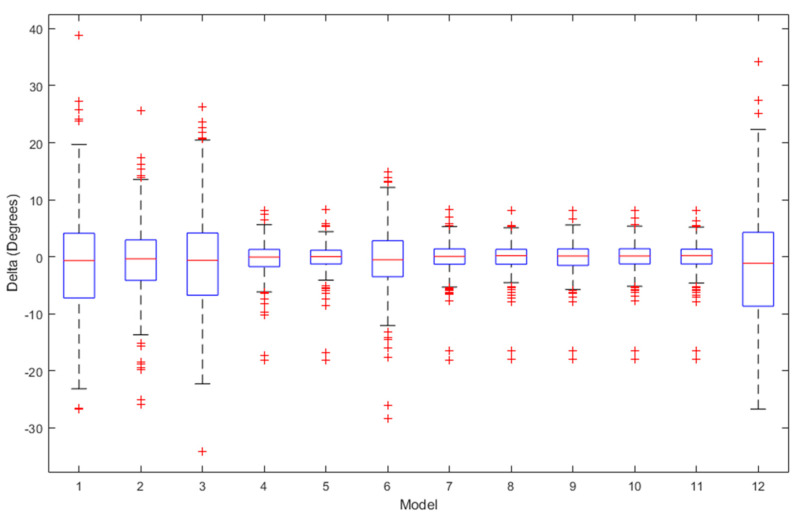
Box plots summarising the hip ROM changes across various regression machine learning algorithms on the test set.

**Figure 11 sensors-24-01337-f011:**
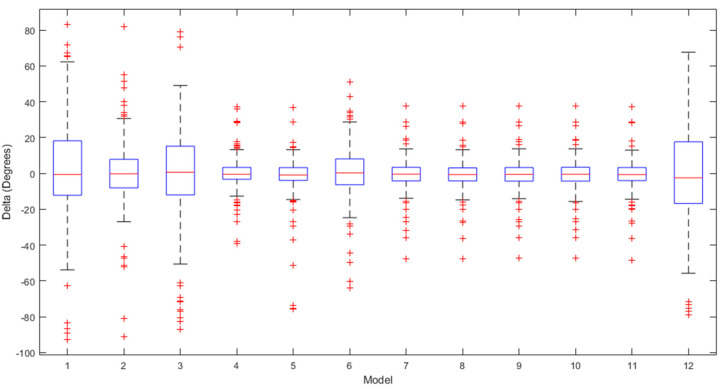
Box plots summarising the knee ROM changes across various regression machine learning algorithms on the test set.

**Figure 12 sensors-24-01337-f012:**
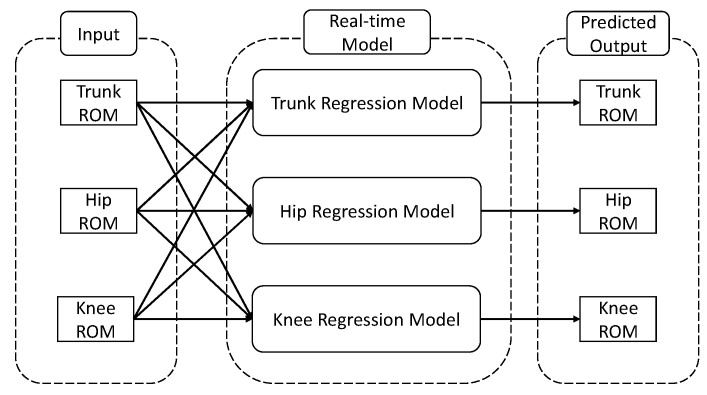
Proposed structure of the real-time prediction model.

**Table 1 sensors-24-01337-t001:** Description of participant demographics.

Variables (Units)	Mean (SD)
Age (years)	45.9 (11.4)
Height (cm)	174.1 (9.8)
Weight (kg)	80.9 (17.5)
BMI (m/kg^2^)	26.7 (5.9)
Pain level (VAS out of 100)	52.5 (17.8)
Duration of pain (months)	60.1 (83.1)
ODI (%)	37.2 (11.2)
PSEQ (out of 60)	44.0 (10.0)

BMI = body mass index; ODI = Oswestry Disability Index; PSEQ = pain self-efficacy questionnaire. VAS = visual analogue scale. SD = standard deviation.

## Data Availability

Restrictions apply to the availability of these data. Data were obtained from the University of Melbourne and are available [from Pranata, A et al. [25]/at https://www.sciencedirect.com/science/article/pii/S0021929018301131, (accessed on 26 July 2020)] with the permission of the University of Melbourne.

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
