# Peer review of "Regression-Based Machine Learning for Predicting Lifting Movement Pattern Change in People with Low Back Pain"

_sensors, 2024, doi:10.3390/s24041337_

Round 1
Reviewer 1 Report
Comments and Suggestions for Authors
The proposal presents a investigation using regression-based machine learning techniques to forecast alterations in trunk, hip, and knee movements subsequent strength training for people have low back pain (LBP). The proposal presents an interesting topic; however, the following aspects were identified:
1. It is suggested to indicate specifically the type of Paper (Article, Review, Communication, etc.) - line 1. After reviewing the proposal, it can be seen that it is of the "Article" type.
2. It is suggested to present more evidence of the participation of the sixty-nine people, i.e., perhaps present the results obtained by the cameras of some of the evaluations of at least 10 participants or other data.
3. It is indicated that a software was developed using LabVIEW 2009 but no further details are provided, so it is suggested to show visual evidence of the software and indicate further details (characteristics of the computer equipment used, structure, functionalities, among other details).
4. It is indicated that the medical staff, based on the results obtained, can observe whether the treatment is effective or not, but how and where can they visualize the results of the patients?
5. At the end of the 12-week treatment, was any study conducted on the patient's perception of the benefit of the treatment? If so, please indicate the results obtained.
Comments on the Quality of English LanguageNA
Author Response
I have attached file for the response to Reviewer 1.

Reviewer 2 Report
Comments and Suggestions for Authors
This paper used various kinds of regression algorithm to predict lifting movement patterns in participants with LBP(low back pain). The authors showed the comparison of the performance of 12 different regression models for trunk, hip, and knee movements.
1. For interested readers, it would be helpful to describe the experimental environment with pictures (or photos).
2. I think a graphical representation of Table 1-3 and Table 4-6 would be helpful to analyze the performance comparatively.
3. For the Gaussian Model, it would be good to give a formula for the applied kernel (specifically, 9, 10, and 11).
4. The authors said they identified four movement patterns in their previous paper [7]. Why did they focus on three joints only in this paper? It would be good to include an explanation of why.
5. In equation (1), ROM was defined as the difference between the maximum and minimum values of rotational displacement. The age range of the participants was very diverse. Was the ROM universally applied to all participants? Or was it applied variably depending on age or pain level? If the MAE of the knees was found to be higher than that of the trunk and hips, wasn't there a significant difference depending on age?
6. The author described that “a motion analysis system consisting of 12 camera”. It would be a good idea to consider extracting image data rather than numerical data and applying a deep learning model such as CNN.
Author Response
I have attached the file in response to Reviewer 2.

Round 2
Reviewer 1 Report
Comments and Suggestions for Authors
The authors correctly addressed the observations of the first review. Now the proposal could be considered for possible publication only by indicating in line 1 the type of work, which is "Article".
Comments on the Quality of English LanguageNA